

# Deciphering regulatory patterns in a mouse model of hyperoxia-induced acute lung injury

Yundi Chen[1,*], Jinwen Liu[2,3,*], Han Qin[4], Song Qin[5], Xinyang Huang[6], Chunyan Wei[7] and Xiaolin Hu[2,6]

[1] Tongji Medical College, Huazhong University of Science and Technology, Wuhan, Hubei, China
[2] School of Public Health, Shanghai Jiao Tong University School of Medicine, Shanghai, China
[3] Department of Nutrition, College of Health Science and Technology, Shanghai Jiao Tong University School of Medicine, Shanghai, China
[4] Department of Respiratory and Critical Care Medicine, Kweichow Moutai Hospital, Zunyi, Guizhou, China
[5] Department of Critical Care Medicine, Affiliated Hospital of Zunyi Medical University, Zunyi, Guizhou, China
[6] Center for Single-Cell Omics, School of Public Health, Shanghai Jiao Tong University School of Medicine, Shanghai, China
[7] Department of Gynecology, Obstetrics and Gynecology Hospital of Fudan University, Shanghai, China
[*] These authors contributed equally to this work.

Corresponding authors
Chunyan Wei,
16111250010@fudan.edu.cn
Xiaolin Hu, 184514@shsmu.edu.cn

## ABSTRACT

**Background**. Oxygen therapy plays a pivotal role in treating critically ill patients in the intensive care unit (ICU). However, excessive oxygen concentrations can precipitate hyperoxia, leading to damage in multiple organs, with a notable effect on the lungs. Hyperoxia condition may lead to hyperoxia-induced acute lung injury (HALI), deemed as a milder form of acute respiratory distress syndrome (ARDS). Given its clinical importance and practical implications, there is a compelling need to investigate the underlying pathogenesis and comprehensively understand the regulatory mechanisms implicated in the development of HALI

**Results**. In this study, we conducted a mouse model with HALI and performed regulatory mechanism analysis using RNA-seq on both HALI and control group. Comprehensive analysis revealed 727 genes of significant differential expression, including 248 long non-coding RNAs (lncRNAs). Also, alternative splicing events were identified from sequencing results. Notably, we observed up-regulation or abnormal alternative splicing of genes associated with immune response and ferroptosis under hyperoxia conditions. Utilizing weighted gene co-expression network analysis (WGCNA), we ascertained that genes involved in immune response formed a distinct cluster, showcasing an up-regulated pattern in hyperoxia, consistent with previous studies. Furthermore, a competing endogenous RNA (ceRNA) network was constructed, including 78 differentially expressed mRNAs and six differentially expressed lncRNAs, including H19. These findings uncover the intricate interplay of multiple transcriptional regulatory mechanisms specifically tailored to the pulmonary defense against HALI, substantiating the importance of these non-coding RNAs in this disease context.

**Conclusions**. Our results provide new insights into the potential mechanisms and underlying pathogenesis in the development of HALI at the post-transcriptional level. The findings of this study reveal potential regulatory interactions and biological roles of specific lncRNAs and genes, such as H19 and Sox9, encompassing driven gene expression patterns, alternative splicing events, and lncRNA-miRNA-mRNA ceRNA

---

networks. These findings may pave the way for advancing therapeutic strategies and reducing the risk associated with oxygen treatment for patients.

# INTRODUCTION

Oxygen therapy is one of the most important interventions in treating critically ill patients, including those with COVID-19, a global pandemic. Oxygen supplementation is the most frequently used therapy to ensure sufficient oxygen supply to organs and tissues. While it is undoubtedly beneficial in the short term, it carries risks in the long term (*Kallet & Matthay, 2013*), including the potential for hyperoxia-induced acute lung injury (HALI), which can eventually lead to acute respiratory distress syndrome (ARDS), a condition with a high fatality rate (*Bellani et al., 2016*; *Fan et al., 2020*). However, only a few effective inventions for ARDS are available to ameliorate patients' pain, including mechanical ventilation with tidal volume, airway pressure (30 cmH$_2$O) limitation, and early prolonged prone-positioning sessions (*Bein et al., 2016*). According to the Berlin definition, ARDS is categorized into three types: mild, moderate, and severe. HALI is considered a mild type of ARDS.

Numerous studies have found that hyperoxia can extensively impair lung function, through damaging vascular endothelium and alveolar epithelium (*Li et al., 2020*). Therefore, a HALI mouse model was utilized to study the pathogenesis of HALI, where mice were exposed to >90% oxygen. Many mechanisms of HALI have been identified using mouse models, including ferroptosis and immune response (*Chen et al., 2022*; *Chen et al., 2021c*; *Soundararajan et al., 2022*; *Yin et al., 2021*). Under high-oxygen conditions, lung cells generate reactive oxygen species (ROS) (*Dias-Freitas, Metelo-Coimbra & Roncon-Albuquerque Jr, 2016*), thus may causing mitochondrial damage, thus disrupting Ca$^{2+}$ homeostasis and damaging mitochondrial respiratory chain (*Soundararajan et al., 2022*), like *Akap1* plays a critical role in the HALI mouse model through mitochondrial dysfunction (*Narala et al., 2018*; *Soundararajan et al., 2022*). ROS can induce alveolar epithelial cells (AEC) to secrete inflammatory cytokines, like IL-1$\beta$, IL-6, IL-8, IP-10, and TGF-$\beta$1 (*Laube et al., 2017*), thus resulting in damage to pulmonary vascular endothelial cells and AECs, destruction of the alveolar-capillary barrier, formation of lung transparent membranes, and further development of pulmonary necrosis and fibrosis, and eventually, leading to ARDS.

Our previous studies demonstrated that gene-gene interactions and post-transcriptional regulation are complex in some diseases (*Hu et al., 2022*; *Wei et al., 2016*; *Zhang et al., 2023*). While numerous studies focus on the destructive role of ROS in cells (*Sies et al., 2022*; *Unsal et al., 2020*), the mechanisms underlying HALI development are complex, involving gene expression patterns, alternative splicing, and competing endogenous RNAs regulation. Therefore, it is worthy and necessary for underlying the complex interactions

and mechanism of regulation of genes involved in the destructive process under ROS and the development of HALI.

With the advancement of next-generation sequencing technologies, transcriptome-wide analysis, including gene expression and alternative splicing, has been available and affordable. One role of long non-coding RNA (lncRNA) in cells is to act as ceRNAs by sponging miRNA with sequencing match, which inhibits miRNA binding to relevant mRNA and triggers degradation (*Salmena et al., 2011*). Recent studies have elucidated the role of various long noncoding RNAs (lncRNAs) in the pathogenesis of HALI, with growing evidence linking lncRNAs to ROS and oxidative stress, and further apoptosis and cell death (*Dutta et al., 2023*). The lncRNA Gadd7 exacerbates mitochondrial dysfunction and apoptosis in alveolar type II epithelial cells by upregulating Mfn1, with knockdown of Gadd7 showing protective effects (*Zhang et al., 2022*). Additionally, lncRNA SNHG6 accelerates lung cell injury *via* the miR-335/KLF5/NF-$\kappa$B pathway, while its knockdown mitigates these effects (*Meng et al., 2022*). FOXD3-AS1 is also implicated in oxidative stress and HALI, with the former regulating apoptosis through miR-150 sponging (*Zhang et al., 2017*). Furthermore, miR-21-5p has been identified as a modulator of HALI, suppressing mitophagy and apoptosis *via* PTEN/AKT pathways (*Qin et al., 2019*). Although lncRNA H19 is not implicated in HALI, it is highly expressed in hyperoxia-induced bronchopulmonary dysplasia (BPD) in newborn mice, where its silencing alleviates pulmonary injury by upregulating miR-17 and downregulating STAT3 expression (*Zhang et al., 2022*), These findings offer a multifaceted perspective on lncRNA-mediated regulation in hyperoxia-induced lung injury.

The present study therefore aimed to underly the potential transcriptional regulation mechanisms and functions of genes for the development of HALI using a hyperoxia-induced mouse model. To this end, a mouse model of HALI was established with hyperoxia condition, and the variation of gene expression pattern was revealed by transcriptome sequencing. Bioinformatics analysis revealed that a total of 727 differentially expressed genes (DEGs), with 248 differentially expressed lncRNAs (DElncRNAs) contained with RNAseqPower being 1.0. Biological processes and pathway enrichment analyses indicated an up-regulated expression pattern in HALI for genes involved in immune response. Furthermore, the alternative splicing analysis found out some genes were not DE but differentially spliced, such as immune response-related genes (*Bax*, *Spink5*, *Gpx2*, *Sox9*, *Il-6*, *Lif*) and ferroptosis-related genes (*Gclc*, *Ftl1*).

Those results indicate that the hyperoxia condition inflicts negative results not only by influencing gene expression but also by affecting alternative splicing. Finally, a lncRNA-miRNA-mRNA regulatory network was constructed in this study. Our finding offers novel insights into the development of HALI and aids in identifying potential therapeutic targets of HALI, which need to be validated by further experiments.

# MATERIALS & METHODS

## Construction of the mouse model

All experimental procedures were performed and approved by Institutional Animal Care and Use Committee of Zunyi Medical University (IACUC; No. zyfy-an-2023-0047),

ensuring compliance with ethical guidelines for animal research. A total of 12 C57BL/6J wild-type mice were purchased from Changsha Tianqin Biotechnology Co., Ltd. (Changsha, China), and randomly assigned into four groups: (i) Control Group ($n = 6$), exposed to zoom air in a regular environment; (ii) HALI group ($n = 6$), where mice were exposed to a hyperoxic environment (5.0 L/min pure oxygen for 72 h), with oxygen levels maintained at $\geq$90% saturation within an air-tight chamber, as previously described (*Liu et al., 2019*). This setup was closely monitored to maintain environmental conditions, including a temperature range of 25–27 °C and relative humidity between 50–70%. The chamber's atmosphere was balanced by using soda lime to absorb the carbon dioxide exhaled by the mice, keeping $CO_2$ levels below 0.5%. The exposure protocol was rigorously scheduled for 23.5 h per day, with a 0.5-hour opening daily for essential maintenance tasks such as providing food, water, and clean bedding, ensuring consistency and replicability of the experimental conditions. No mice showed any signs of adverse weight loss in the two groups, and therefore no mice were excluded from the study. All experimental animals were anesthetized using isoflurane inhalation and subsequently euthanized, ensuring a painless end. Of note, euthanasia of animals strictly followed a protocol approved by our IACUC and the American Veterinary Medical Association. Animals needed for H&E staining were euthanized as described below (Tissue Collection and H&E staining procedure). To the further extent possible, our manuscript followed the ARRIVE guidelines (Checklist S1) (*Percie du Sert et al., 2020*).

## Tissue collection and H&E staining procedure

On the day of sacrifice, mice exposed to 90% oxygen for 72 h in a hyperoxic chamber were euthanized using an intraperitoneal injection of pentobarbital (50 mg/kg). After the completion of euthanasia, meticulous disinfection and preparatory measures were undertaken to ascertain the sterility of both the surgical field and the instruments employed. A representative lung tissue sample was typically procured post-thoracotomy, with the chosen site for this specific study being the lower lobe of the right lung. Lung tissues from the mice exposed to hyperoxia were then harvested and processed into sections on glass slides. These slides were deparaffinized in xylene and rehydrated through a series of graded alcohols (100%, 90%, and 70%). Subsequently, the sections underwent staining with hematoxylin for 3 min, followed by eosin staining for 2 min, and were then rinsed in distilled water. Finally, the sections were dehydrated through a second series of graded alcohols (70%, 90%, and 100%), cleared in xylene, and evaluated for lung injury using a light microscope.

## RNA isolation and quality control

Lung tissues were collected from the mice and processed accordingly. Total RNA was extracted using TRIzol reagent (Magen, Guangzhou, China) following the manufacturer's instructions. RNA quantity and purity were evaluated spectrophotometrically using a Nanodrop ND-2000 (Thermo Fisher Scientific, Waltham, MA, USA) based on the A260/A280 ratio. RNA integrity was further assessed using an Agilent Bioanalyzer 4,150 (Agilent Technologies, Santa Clara, CA, USA), and only RNA samples with an RNA integrity number (RIN) $\geq$8 was used for library construction.

## Library preparation for RNA-seq

Library preparation was performed following the Abclonal mRNA-seq Library Preparation Kit (Abclonal, Wuhan, China) protocol. For each sample, 1 µg of total RNA was used to enrich and purify polyadenylated mRNAs using oligo (dT) beads. The purified mRNAs were then reverse transcribed into cDNA using reverse Rnase H and DNA polymerase I. The cDNA libraries were validated for appropriate size distribution using an Agilent Bioanalyzer 4150 (Agilent Technologies). Finally, the libraries were subjected to 150 bp paired-end sequencing on an Illumina NovaSeq 6,000 platform.

## Quality control and read mapping

The mouse reference genome and gene annotation files (GRCm39, release M32) were downloaded from the GENCODE database (*Frankish et al., 2021*). Adapter sequences, low-quality reads (≥20% of bases with Phred score <15), and reads containing >5% ambiguous bases were trimmed using Trim Galore (v0.6.4) (*Krueger, 2015*). The quality of the trimmed reads was assessed using FastQC (v0.12.0) (*Andrews, 2010*). The cleaned reads were then aligned to the reference genome using STAR (v2.7.10b) (*Dobin et al., 2013*) with default parameters. Read counts were derived from the alignments and quantified at the gene level. In total, 12 samples were included and are evenly separated into 2 groups, with 6 biological replicates on each side. The statistical power of this experimental design, calculated in RNASeqPower (*Hart et al., 2013*) is 1.0.

## Data deposit and acquisition

The datasets generated during this investigation, are meticulously deposited in the Gene Expression Omnibus (GEO) (*Clough & Barrett, 2016*). Processed gene counts from this study that are directly downloadable from the GEO website (https://www.ncbi.nlm.nih.gov/geo/query/acc.cgi?acc=GSE237260).

The raw sequencing data, initially captured as fastq files, are accessible through the SRAToolkit (*Sayers, O'Sullivan & Karsch-Mizrachi, 2022*), enabling the conversion to full-length, paired-end sequence reads (fastq format).

## Identification of differentially expressed genes

Gene expression for each sample was consolidated using ensemble IDs, and differential expression analysis was executed with DESeq2 (v1.36.0) (*Love, Huber & Anders, 2014*). Significant differentially expressed genes were identified based on the following criteria: $\log_2$ fold change >1.5 or <−1.5 and *P*-values <0.05. Visualization of differentially expressed genes was accomplished using a volcano plot generated with the ggplot2 package (v3.3.6) (*Wickham, 2011*) and a heatmap created with the ComplexHeatmap package (v2.10.0) (*Gu, Eils & Schlesner, 2016*) in R 4.1.0 (*Ihaka & Gentleman, 1996*).

## GO, KEGG and WikiPathways enrichment analyses

Gene Ontology (GO) (*Ashburner et al., 2000*), KEGG pathway (*Kanehisa & Goto, 2000*), and WikiPathways (*Slenter et al., 2018*) enrichment analyses were conducted to examine the biological functions of differentially expressed genes (DEGs). These genes were categorized into two groups according to their regulatory directions (up- or down-regulated), based

on log$_2$ fold change values. Enrichment analysis was performed and visualized using the ClusterProfiler package (version 4.2.2) (*Wu et al., 2021*) in R software (version 4.1.0) (*Ihaka & Gentleman, 1996*). Terms with *P*-values less than 0.05 were considered significantly enriched.

## Construction of WGCNA network

Differentially expressed genes were clustered into distinct groups using the WGCNA package (v1.70.3) (*Langfelder & Horvath, 2008*) in R, according to their expression patterns. Modules exhibiting a module significance score <0.05 were identified as recurrence-associated modules.

Similarly, distinct gene clusters were subjected to GO and KEGG pathway enrichment analysis using the ClusterProfiler package (v4.2.2) (*Wu et al., 2021*) to identify various aspects of biological functions. Terms with a *P*-value <0.05 were considered significantly enriched.

## Identification of differential alternative splicing events

Splicing is an essential process that determines various cellular fates and directs diverse biological pathways. In this study, we utilized mapped read data from each sample (in BAM format) and combined it with mouse annotation files (in GTF format) as input. We employed rMATS (v3.3.0) (*Shen et al., 2014*) to quantify the splicing events for each sample and identify differential alternative splicing events

A statistical summary of alternative splicing for each sample was conducted using the summary.py script from rMATS (v4.1.2) (*Shen et al., 2014*). Differential alternative splicing events with a *P*-value <0.05 were considered significant. If a gene exhibited more than one type of alternative splicing, the most significant event was selected for visualization in a volcano plot using ggplot2 (v3.3.6) (*Wickham, 2011*) in R 4.1.0 (*Ihaka & Gentleman, 1996*).

## Exploring the lncRNA and mRNA targets of miRNAs

There are lots of tools and databases available for predicting or collecting miRNA-target gene interactions. MultiMir (*Ru et al., 2014*) is one of the most comprehensive databases, incorporating both validated miRNA-target interaction databases (miRecords (*Xiao et al., 2009*, miRTarBase (*Huang et al., 2020*), TarBase *Sethupathy, Corda & Hatzigeorgiou, 2006*)) and predicted miRNA-target interaction databases (DIANA-MicroT-CDS (*Paraskevopoulou et al., 2013*), EIMMo, MicroCosm, miRanda, miRDB, PicTar, PITA, and TargetScan *McGeary et al., 2019*). Moreover, it provides an R package to enable user-friendly downloading of these results. In this study, we utilized MultiMir (version 1.22.0) (*Ru et al., 2014*) to query DEGs and differentially expressed long non-coding RNAs targeting miRNAs, and subsequently saved the obtained results.

## Construction of ceRNA network

According to the ceRNA hypothesis, differentially expressed mRNA, differentially expressed lncRNA, and their corresponding targeting miRNAs were collected using the multiMiR (v1.22.0) (*Ru et al., 2014*) package in R (v4.1.0) (*Ihaka & Gentleman, 1996*). The lncRNA-miRNA-mRNA ceRNA network was constructed based on these results and visualized using Cytoscape (v3.10) (*Shannon et al., 2003*).
## RESULTS

### Generation of mouse model of HALI

A model of HALI was developed utilizing twelve C57BL/6J mice (Figs. 1A, 1B). Six mice were randomly selected and exposed to hyperoxia ($\geq 90\%$ oxygen) within an environmentally regulated chamber (25 °C–27 °C, 50–70% humidity) for 72 h (details in Methods). The remaining six mice were housed in an identical environment with standard oxygen concentration, serving as the control group. Hematoxylin and eosin (H&E) staining was conducted to verify the presence of HALI-associated pathological features. As depicted in Fig. 1B, exposure to hyperoxia resulted in considerable lung injury, characterized by alveolar capillary barrier disruption, pulmonary edema, and infiltration of inflammatory cells. These observations were identical to the clinical observation of the HALI (*Kallet & Matthay, 2013*), thereby confirming the successful establishment of the HALI murine model.

### Identification of differentially expressed genes

The RNA-seq data from six mice with HALI and six control mice were analyzed for a comprehensive identification of genes and isoforms associated with HALI. Gene expression patterns of each sample were quantified with STAR (*Dobin et al., 2013*), and differential analysis was conducted utilizing DESeq2 (*Love, Huber & Anders, 2014*). A total of 727 differentially expressed genes (DEGs) were identified (Table S1), with genes exhibiting a $\log_2$ fold change $>1.5$ or $<-1.5$ and the $P$-value $<0.05$ considered as significantly DEGs. To validate our analysis parameters, RNAseqPower was employed with a depth of 413, alpha of 0.1, effect size of 727/56953, and a sample size of 6, yielding an RNAseqPower of 1. This confirms the suitability of our parameters for DEGs identification in this dataset. Among these DEGs, 208 were up-regulated in the HALI group (including 43 lncRNAs) and 519 were down-regulated (including 205 lncRNAs) (Figs. 2A, 2B). Notably, genes associated with HALI were also detected differentially in this study (Table S1), such as *Spink5*, *Gpx2*, *Lif*, and *Ear1*, which are related to immune processes (*Ahmed et al., 2022*; *Bonfill-Teixidor et al., 2024*; *Chen et al., 2024*; *O'Reilly et al., 2012*) and, like *Bax*, may lead to cell death through the apoptosis process (*Budinger et al., 2011*). *Gclc* and *Ftl1* are associated with ferroptosis (*Luo et al., 2022*; *Wang et al., 2021*).

### Determination of the biological functions of DEGs

To address the biological functions of the DEGs, we conducted enrichment analyses of Gene Ontology (GO) (*Ashburner et al., 2000*), Kyoto Encyclopedia of Genes and Genomes (KEGG) (*Kanehisa & Goto, 2000*), and WikiPathways (*Slenter et al., 2018*) *via* the ClusterProfiler (*Wu et al., 2021*) package in R (Table S2). The up-regulated DEGs in HALI were predominantly enriched in immune response-related functions, such as cellular response to interleukin-1, ERK1 and ERK2 cascade, and monocyte chemotaxis, within the biological process (BP) category of GO (Fig. 2C). Conversely, down-regulated DEGs (including *Actn2*, *Agt*, *Casq2*, *Dhrs7c*, *Drd4*, *Grin1*, *Hamp*, and so on) in HALI were connected to muscle cell development and myofibril assembly (Fig. 2C).

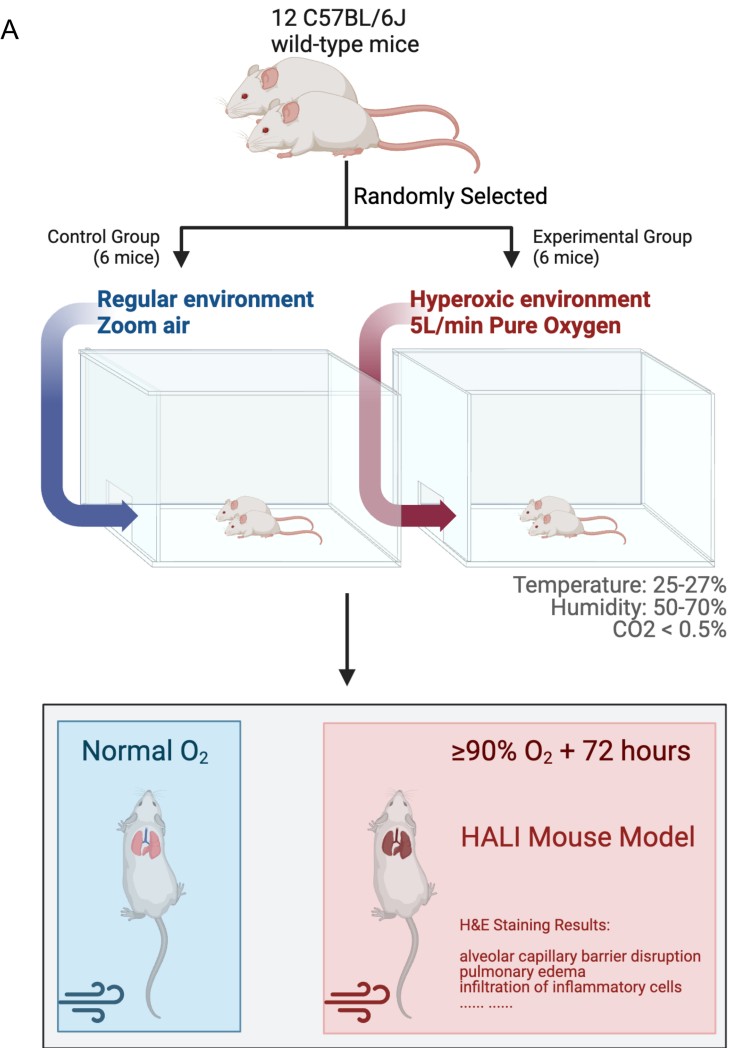

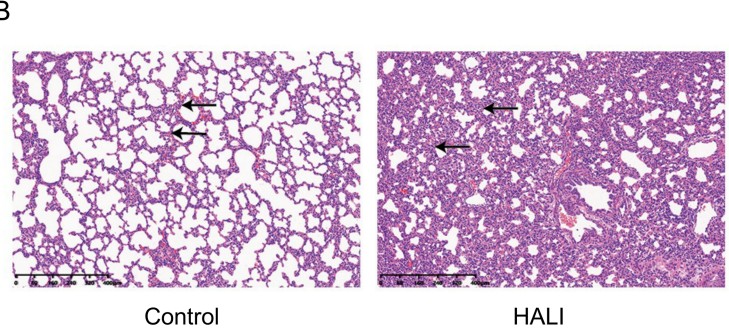

**Figure 1** **Establishing and validating mouse model of Hyperoxia-indeced Acute Lung Injury (HALI).**
(A) Pattern diagram for constructing mouse model of HALI (created with Biorender.com). (B) H&E staining of lung tissue in the mouse model of healthy control (left) and HALI (right).

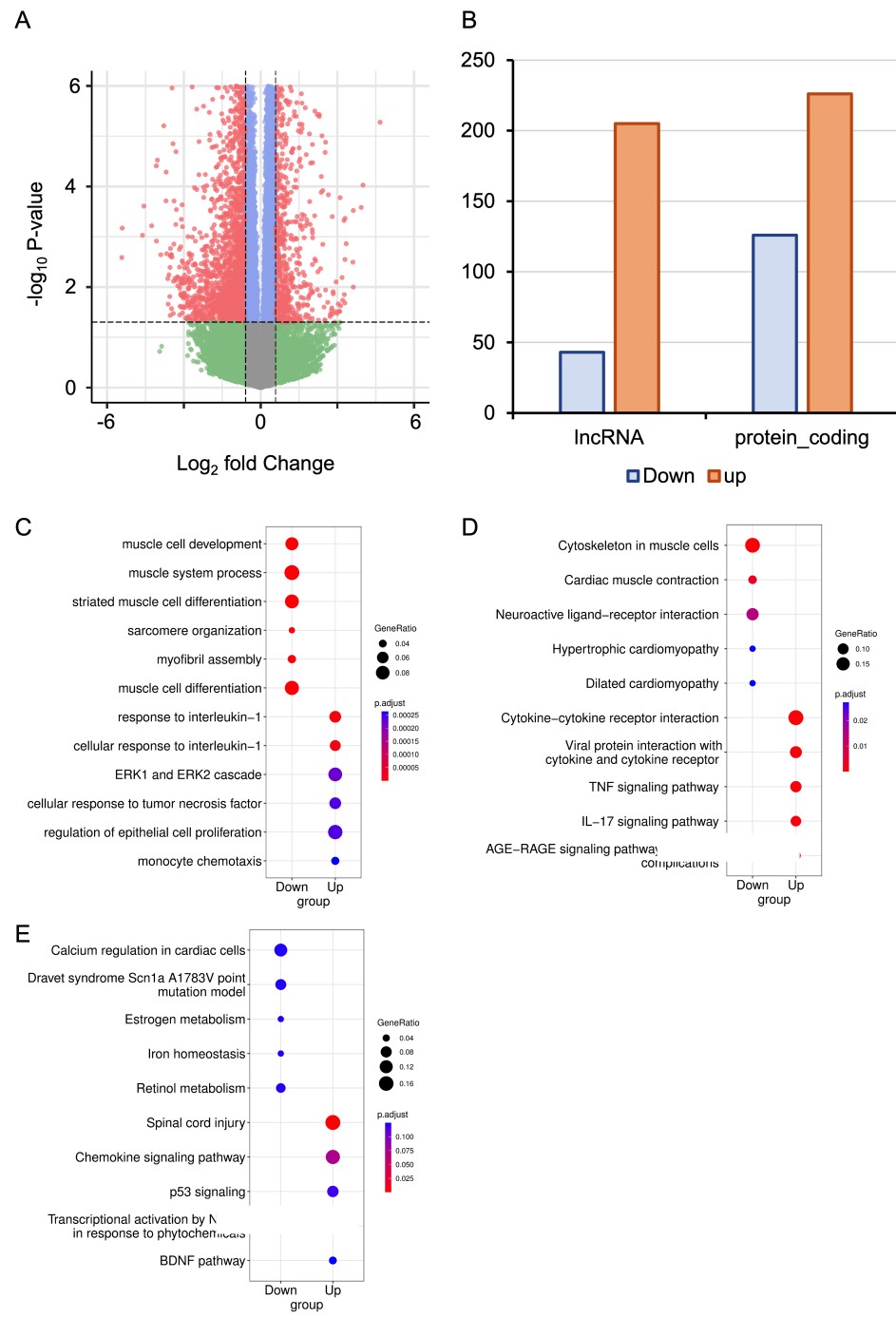

**Figure 2 RNA-seq analysis of differentially expressed Genes (DEGs) in hyperoxia-induced acute lung injury (HALI).** (A) Volcano plot of DEGs between the model of HALI and the control group. (B) Difference between up-regulated and down-regulated DEGs in mRNA and lncRNA. (C–E) Enrichment analysis for differentially expressed genes, including Gene Ontology (GO) enrichment analysis in biological process (C), Kyoto Encyclopedia of Genes and Genomes (KEGG) pathway enrichment analysis (D), and the enrichment results of WikiPathways (E).

The results of the KEGG pathway enrichment analysis demonstrated that up-regulated DEGs were associated with cytokine-cytokine receptor interactions, tumor necrosis factor (TNF) signaling pathway, and interleukin-17 (IL-17) signaling pathway (Fig. 2D). Conversely, down-regulated DEGs were implicated in biological processes such as cardiac muscle contraction, neuroactive ligand–receptor interaction, and dilated cardiomyopathy, among others (Fig. 2D).

WikiPathways (*Slenter et al., 2018*) is an open, continuously updated, and curated pathway database that encompasses 202 pathways for *Mus musculus*, covering more than 4,500 genes (version 20230610). In this study, we utilized this database to augment the pathway resources of the KEGG database. The up-regulated genes were primarily linked to the chemokine signaling pathway, P53 signaling, and brain-derived neurotrophic factor (BDNF) pathway. In contrast, down-regulated genes were correlated with calcium regulation in cardiac cells, iron homeostasis, and so forth (Fig. 2E).

Given that the DEGs participated in multiple biological functions, it was crucial to discern the genes exhibiting distinct patterns to identify the complex biological processes involving these DEGs.

## Construction of WGCNA network

Weighted gene co-expression network analysis (WGCNA) is a hierarchical clustering approach employed to categorize genes into discrete expression patterns. In our study, we utilized WGCNA to identify five distinct clusters of DEGs, employing a soft thresholding power of 14 and applying module filtering with specific cut-off scores (Fig. 3A, Table S3). Among these clusters, the G1 module encompassed merely four genes, and due to its score falling below the significance threshold, it was filtered out with subsequent analyses. The G5 module demonstrated a positive association with the HALI cohort while the G3, G4, and G2 modules displayed negative correlations (Fig. 3A). The G3, G5, and G4 modules comprised approximately 200 genes each. The G2 module, on the other hand, contained a minor fraction of DEGs (56 genes) (Fig. 3B).

The genes from various modules were subjected to GO and KEGG enrichment analyses. The enrichment results demonstrated that genes in distinct groups were involved in diverse biological processes (Figs. 3C, 3D, Table S4). Genes within the G5 module were correlated with immune response (*e.g.*, response to interleukin-1, ERK1 and ERK2 cascade), which resembled the up-regulated genes' enriched terms in DEG analysis. Genes within the G4 module were related to muscle system processes, muscle cell development, myofibril assembly, and so forth. Meanwhile, genes in the G2 module were associated with G1 to G0 transition, cardiac cell fate commitment, and other relevant processes. Genes in the G3 module were associated with alcoholism, systemic lupus erythematosus.

Upon examination of the enrichment analysis, it appears that genes exhibiting down-regulation in response to HALI events can be grouped into two distinct patterns. Conversely, up-regulated genes display a strikingly similar pattern across instances.

## Identification of differential alternative splicing

Alternative splicing (AS) expands the number of proteins and is a key post-transcriptional regulatory mechanism. Therefore, investigating AS events is crucial for understanding the

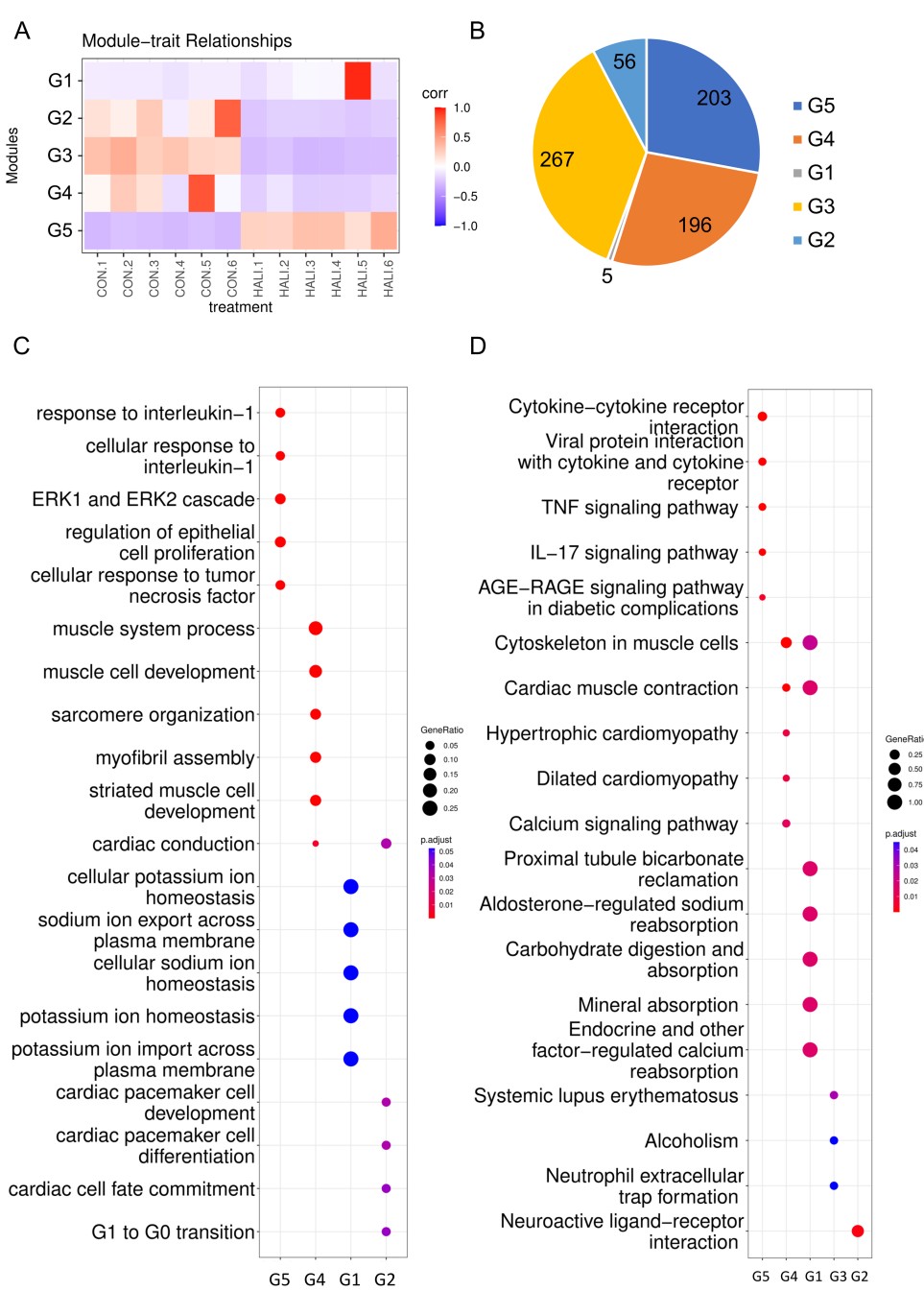

**Figure 3** **WGCNA of differentially expressed genes (DEGs).** (A) Hierarchical clustering heatmap depicting the co-expressed genes in different modules. The rows represent different samples, while the gene expression patterns of the different groups are depicted in separate columns. (B) Number of DEGs in the different groups. Results of (C) GO and (D) KEGG enrichment analysis of the DEGs in the different groups.

development of HALI. To this end, we employed rMATS to detect five types of AS events: skipped exon (SE), alternative 3′ splicing sites (A3SS), alternative 5′ splicing sites (A5SS),

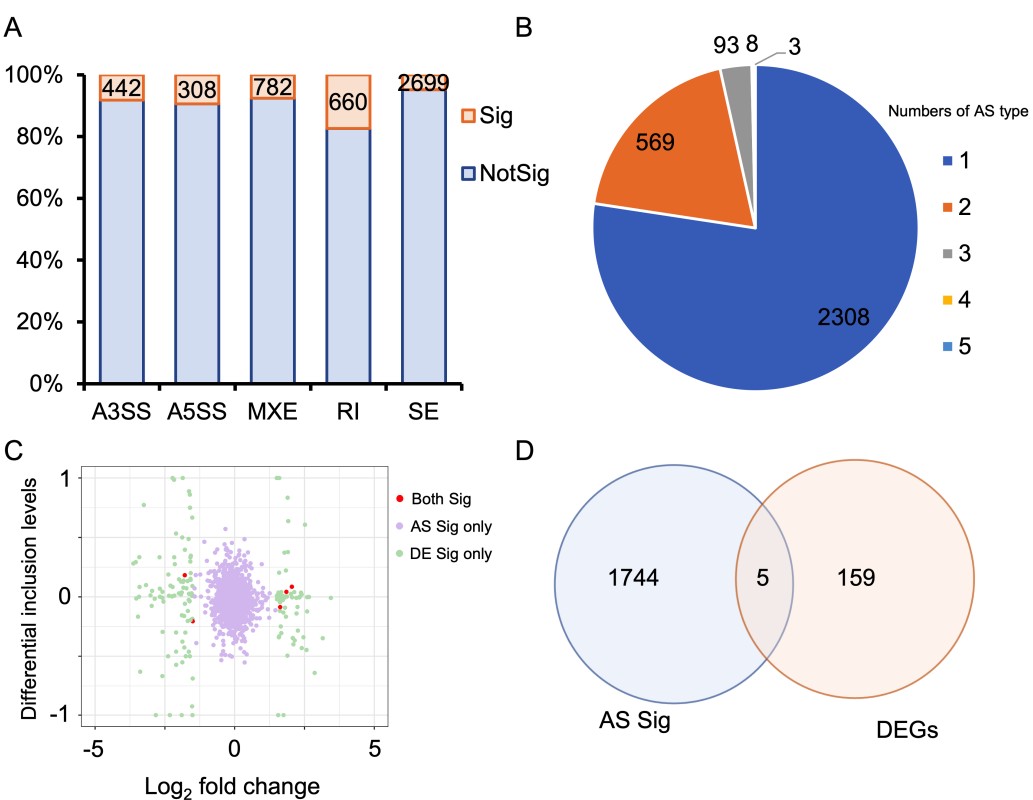

**Figure 4** **Alternative splicing (AS) events between HALI and control group.** (A) Numbers of significant (Sig) and not significant (NotSig) for different types of alternative splicing events. Events with $P$-value $< 0.05$ are treated as significant. (B) Numbers of types of significant AS events for specific genes. (C) Comparison of differentially expressed (DE) results and alternative splicing (AS) events. The red dot means both significant (Both Sig), the purple dot indicates significant AS but not DE (AS Sig only), and the green dot indicates significant DE but not AS (DE Sig only). (D) The intersection of significant AS (AS Sig) and significant DE (DE Sig).

mutually exclusive exons (MXE), and retained intron (RI) (*Shen et al., 2014*). AS events with a $P$-value $<0.05$ were considered as differentially alternative splicing events (Table S5). For specific genes exhibiting multiple splicing events, the AS events with the minimum $P$-value and higher inclusion level difference were selected as the typical differential AS events (Table S6). As depicted in Fig. 4A, approximately 10% of AS events were differential. SE was the predominant AS type, followed by MXE, RI, A3SS, and A5SS, which had the fewest differential AS events. Most genes with differential AS exhibited only one type of AS. Notably, only three genes—*Gas5*, *Smox*, and *Snhg17*—demonstrated all five types of differential AS events (Fig. 4B). *Gas5*, a long non-coding RNA (lncRNA), functions as a competitive endogenous RNA (ceRNA) that sequesters *miR429*, thereby inhibiting *Dusp1* in HALI (*Li & Liu, 2020*). *Smox* expression has been reported to increase under hyperoxia-induced neuronal damage in the retina (*Narayanan et al., 2014*).

Regarding genes not significantly expressed but exhibited differential alternative splicing (AS), we juxtaposed our DEGs with differentially AS events (Table S7). 159 genes showed

differentially expressed but not AS, 1,744 genes showed AS but not differentially expressed, and only five genes showed both significance in AS events and DE genes (Asns, Lrrc74b, Nr4a1, Slc26a10, Snhg15) (Figs. 4C, 4D). Among those five genes, Snhg15 has been implicated in the suppression of cell apoptosis, expression of inflammatory cytokines, and oxidative stress response by up-regulating miR-362-3p expression in IPS-induced vascular endothelial cell (*Liu et al., 2022*). Overexpression of *Nr4a1* could reduce the damage of *Dex* on hypoxia reoxygenation-induced in mouse pulmonary vascular endothelial cells (*Dong et al., 2020*).

## Alternative splicing events involved in immune response and ferroptosis

Immune response plays a crucial role in the development of HALI (*Chen et al., 2022*; *Soundararajan et al., 2022*). Therefore, analyzing alternative splicing events of immune-related genes is of particular interest. To this end, we surveyed a list of immune genes from the Gene Ontology database (immune system process, GO:0002376), including 2,929 genes.

We discovered that 60 genes exhibited differential expressions during the HALI onset (Fig. 5A, Table S8). By intersecting these results with AS results from rMATS (*Shen et al., 2014*), we identified 1,237 events with significant alternative splicing when HALI developed, covering 422 genes. Our findings revealed those immune-related genes underwent alternative splicing during HALI (Fig. 5B). Notably, we observed a significant up-regulation of *Bax* during HALI (Fig. 5A), concurrent with significant AS happened to Tmbim6. Given in previous studies that *Tmbim6* acts as a suppressor of *Bax* (*Seitaj et al., 2018*), the abnormal AS of *Tmbim6* might account for the up-regulation of *Bax*.

Nearly half of the AS events were classified as skipped exons (SE), while only a small portion of AS events were categorized as alternative 3′ splicing sites (A3SS) and alternative 5′ splicing sites (A5SS) (Fig. 5C). Most genes harbored only one type of AS event; however, some genes exhibited multiple AS types. For example, *Adgrf5*, a gene associated with airway inflammation and potentially regulating CCL2-mediated inflammation (*Kubo et al., 2019*), harbored four types of AS events (excluding RI), and another gene, *Lilrb4b*, featured four types of AS events (excluded MXE) (Fig. 5D).

In addition to immune response, ferroptosis is one of the key mechanisms in the occurrence of HALI (*Yin et al., 2021*). We retrieved a list of 33 ferroptosis-related genes from the KEGG database (KEGG ID: map04216). A substantial number of these genes exhibited up-regulation in the context of HALI (Fig. 6A, Table S9), albeit not statistically significant, corroborating previous findings. Intriguingly, several genes demonstrated significant alternative splicing despite the absence of significant differential expressions, such as *Gss*, *Trp53*, and *Lpcat3* (Fig. 6B). This observation underscores the intricate and precise regulatory mechanisms at the transcriptional level.

## Identification of miRNA-target genes and lncRNAs

A total of 248 differentially expressed lncRNAs were identified in HALI, including *H19* (Table S1). This finding prompted us to investigate the potential functions of these

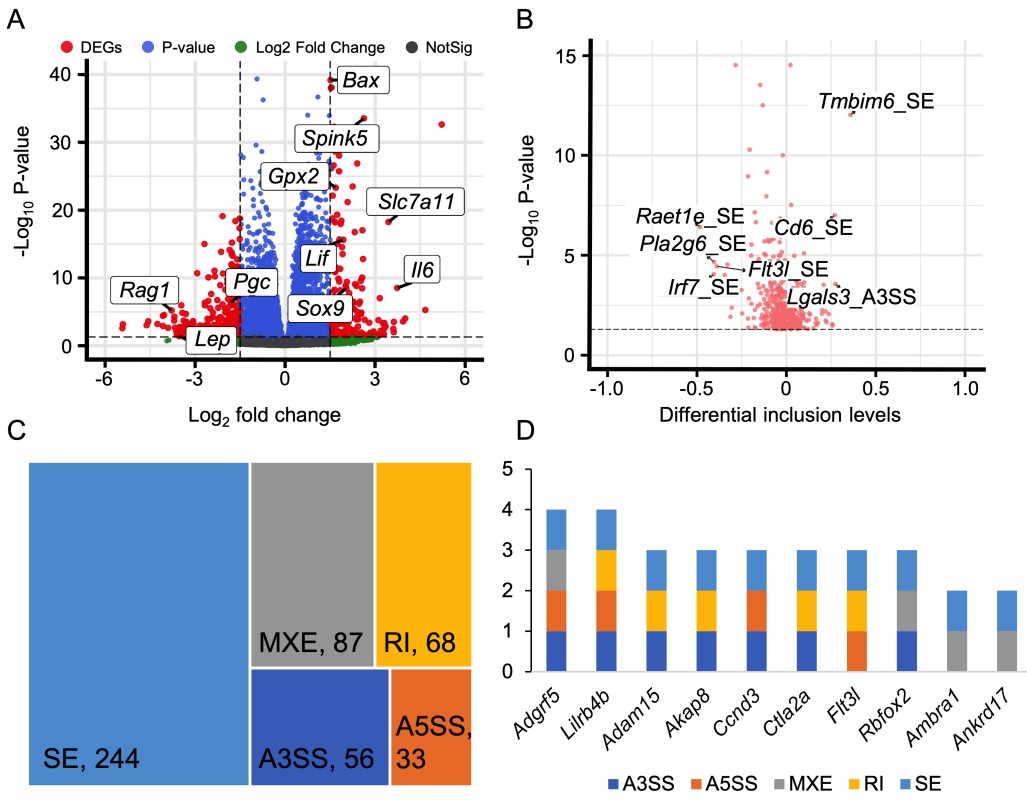

**Figure 5 Alternative splicing events in immune response.** (A) Volcano plot for immune-associated genes. (B) Volcano plot for AS events of immune-associated genes. *X* axis indicated the inclusion level between HALI and control group. Label for the typical gene is a combination of gene symbol, AS type and reads type (JC indicates junction reads only, details described by rMATS). (C) Percentage of genes in different AS types. (D) Numbers of AS types for typical genes.

lncRNAs. One known function of lncRNAs is their role as molecular sponges for miRNAs. By binding to miRNAs, lncRNAs can prevent miRNAs from degrading their target mRNAs. (*Salmena et al., 2011*). MiRNA target genes were obtained from the multiMiR (*Ru et al., 2014*) database and annotated as either predicted or experimentally validated (Table S10). Not surprisingly, the majority of lncRNAs harbored one single miRNA targeting a specific mRNA, while *Gm10447* harbored 16 miRNAs targeting *Cacna2d2*, as well as an additional 12 miRNAs targeting another gene, *Has2* (Fig. 7A).

A comprehensive lncRNA-miRNA-target gene (ceRNA) network was constructed through the integration of differential analysis and miRNA target investigation (Fig. 7B, Table S10). Examination of the ceRNA network disclosed potential biological functions of specific lncRNAs, such as *H19*, *GM10382*, *GM10447*, and *GM10244*. Furthermore, it unveiled potential regulatory interactions of HALI-associated genes, including *Slc7a11*, *Sox9*, *Mfsd2a*, *Nr1d1*, *Ptgs2*, and *Agt*. These genes were implicated in the modulation of cellular ketone metabolic processes (GO:0010565), which is pivotal in the pathogenesis of HALI. Collectively, the ceRNA network analysis elucidated the putative roles of several

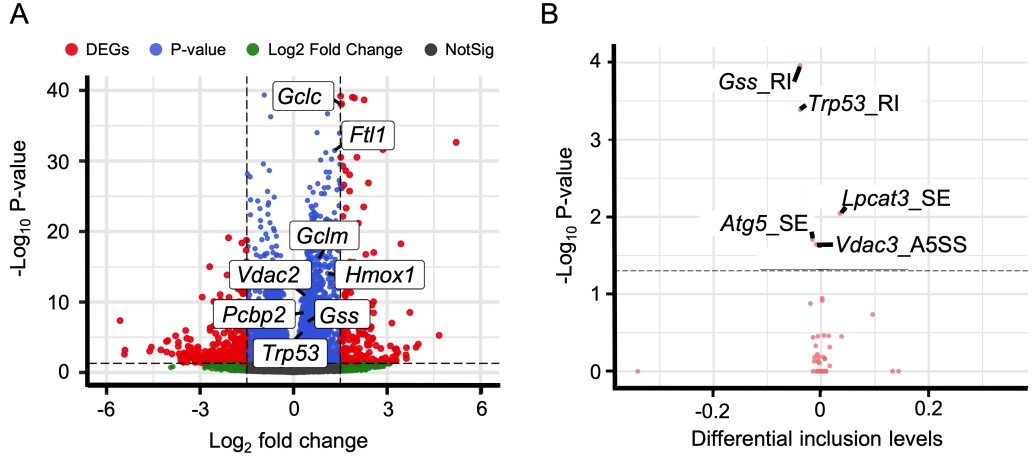

**Figure 6** **Alternative splicing events involved in ferroptosis.** (A) Volcano plot for immune-associated genes. (B) Volcano plot for AS events for immune-associated genes. *X* axis indicated the inclusion level between HALI and control group. Label for the typical gene is a combination of gene symbol, AS type and reads type (JC indicates junction reads only, details described by rMATS).

lncRNAs and genes in the development of HALI, providing valuable insights for future investigations.

## DISCUSSION

HALI is a mild form of ARDS as well as a severe outcome of oxygen therapy. Several previous studies have found that mice exposed to >90% oxygen can develop HALI. To investigate the potential mechanisms of HALI development, a mouse model of HALI was established in this study. 12 C57BL/6J mice were purchased, six mice were randomly selected as the HALI group and housed in a hyperoxia (>90% oxygen) chamber for 72 h, and others were severed as the control group (details in Method). Then the lung tissues were harvested from these mice and wild-type mice. As depicted in Fig. 1B, tissues from hyperoxia group exhibited cellular death, a typical phenomenon of HALI observed in humans. These observations validate that the constructed mouse HALI model was suitable for studying the biological processes and transcriptional regulation mechanisms underlying HALI development.

High oxygen concentration is disastrous for lung cells, triggering a series of destructive processes. In the tissue and organ level, injury induced by Hyperoxic toxicity includes damaged pulmonary capillary endothelium, alveolar type I epithelial cell death, type II epithelial cell hypertrophy, interstitial edema, neutrophil accumulation, altered surfactant production, and decreased lung compliance (*Amarelle et al., 2021*). Initially, reactive oxygen species (ROS) is elevated abnormally (*Kim et al., 2014*) induced by the activation of NOX enzymes, and then causes oxidative stress extensively, represented by lipid peroxidation (*Peterson et al., 2020*), oxidative DNA damage (*Kundumani-Sridharan et al., 2019*), and protein carbonylation (*Paulsen & Carroll, 2013*). Then, ROS-induced cell death happened through intrinsic and extrinsic pathways, including apoptosis, pyroptosis, ferroptosis,

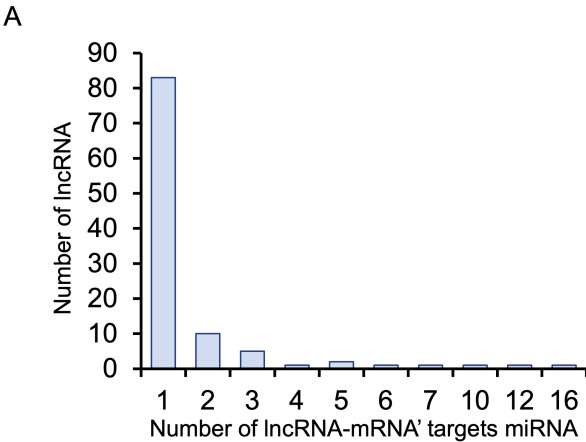

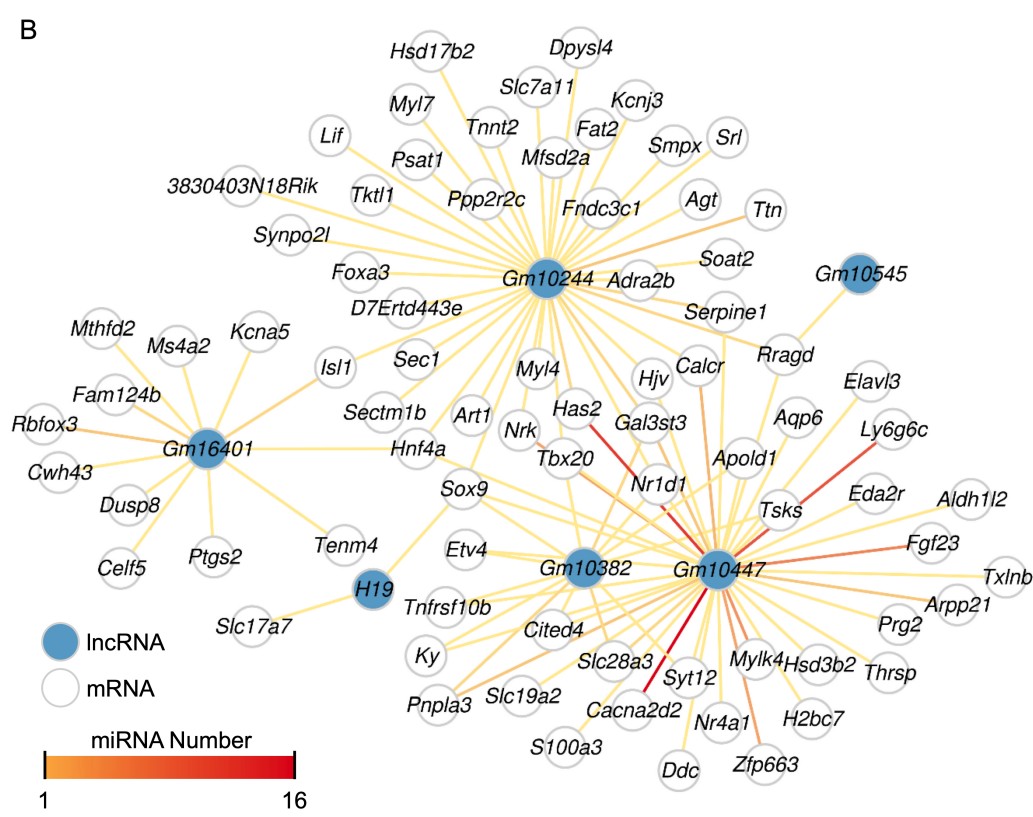

**Figure 7  The ceRNA network for key genes.** (A) Number of miRNAs targeting lncRNA-mRNA pairs. (B) The ceRNA network. The blue dot indicates lncRNA. Deeper red lines indicate the involvement of a higher number of miRNAs.

and necroptosis (*Sharma & Flora, 2021*; *Alva et al., 2023*). Severe HALI, with long-term exposure to high oxygen concentration, may cause lung tissue fibrosis, and ultimately turn into ARDS. Obviously, this is a destructive process under hyperoxia conditions, and previous studies suggest genetic factors involved in HALI development (*Liu et al., 2022*).

While previous studies focused on genes associated with HALI development at the expression level, the mechanisms underlying post-transcriptional regulation have not been extensively studied. To address the comprehensive mechanisms of HALI development, we performed transcriptome sequencing of 12 mouse lung tissues (6 HALI mouse models and 6 controls).

The hematoxylin and eosin H&E staining results of the present study are highly consistent with the results in previous reports (*Kallet & Matthay, 2013*), confirming the reliability of our mouse model to investigate the mechanisms underlying gene transcriptional regulation (Fig. 1). Then, gene expression differences between the HALI mice and healthy controls were analyzed, with 727 DEGs identified. Next, WGCNA was employed for clustering the DEGs exhibiting similar expression patterns. A total of five DEG modules were obtained *via* WGCNA. The G1 module, containing only five genes, was filtered out by WGCNA. DEGs in the different modules are involved in completely different biological processes (Figs. 3C, 3D). DEGs in the G5 module were associated with immune response, such as response to IL-1, the ERK1 and ERK2 cascade, along with TNF signaling pathway. Immune responses are considered a significant process during the development of HALI, as dying cells can stimulate the release of inflammatory cytokines and trigger innate and/or adaptive immune responses (*Hong et al., 2021*; *Patel et al., 2006*). Inflammatory cells such as polymorphonuclear leukocytes generate ROS that can lead to the death of alveolar epithelial cells (*Entezari et al., 2014*). In turn, epithelial cells secrete proinflammatory cytokines which disrupt the alveolar-capillary barrier (*Dias-Freitas, Metelo-Coimbra & Roncon-AlbuquerqueJr, 2016*; *Reddy, Hassoun & Brower, 2007*). Treatment with interleukin-1 receptor antagonist reduced other interleukin and cytokine levels downstream of IL-1 $\beta$ signaling, thereby preventing alveolar disruption and airway fibrosis *in vivo* (*Royce et al., 2016*) (Figs. 3C, 3D). DEGs in the G4 module were primarily enriched for processes related to muscle cell development, myofibril assembly (Figs. 3C, 3D). This initial observation of genes associated with muscle in lung cells may relate to alveolar septum thickening under hyperoxia conditions (Fig. 1) and previous reports of collagen deposition and pulmonary fibrosis in rodents (*Chen et al., 2022*). For example, *P2rx2* enriched in those terms was reported to be up-regulated under intermittent hypoxia in rat lungs by quantitative real-time PCR (*Wu et al., 2008*). DEGs in the G2 module were associated with G1 to G0 transition, cardiac pacemaker cell differentiation.

Many studies have also investigated the roles of gene regulatory mechanisms in the progression of HALI. Alternative splicing (AS) is a key regulatory mechanism that enables limited genes to be translated into numerous proteins. Previous studies have suggested the importance of AS in HALI pathogenesis, with genes involved in mRNA splicing found to be up-regulated under hyperoxia (*Tiboldi et al., 2022*). Other studies have shown that *Il-13* selectively stimulates certain *Vegf164* isoforms, and the combination of *Il-13* and hyperoxia can increase the expression of other *Vegf120* and *Vegf188* isoforms in HALI (*Corne et al., 2000*). However, a systematic and comprehensive analysis of AS during HALI remains to be conducted. To bridge this gap in knowledge, we conducted a comprehensive analysis of AS during the development of HALI. Our analysis identified a total of 1,237 AS events across 422 genes. We also focused on two major processes of HALI pathogenesis: immune

response and ferroptosis. Our results revealed that some genes exhibited significant alternative splicing, even if they were not significantly expressed. For example, *Cd6* is a lymphocyte surface marker involved in TCR signaling in non-small cell lung cancer (*Moreno-Manuel et al., 2020*) and has three isoforms. Additionally, *Trp53*, a gene encoding tumor protein 53 and involved in ferroptosis, was not significantly expressed in HALI but showed one retained intron (RI) event. This gene induces cell cycle arresting, apoptosis, senescence and other processes, and has two isoforms. By analyzing these alternative splicing events, we aim to deepen understanding of the immune response's role in HALI development and potentially identify novel therapeutic targets.

LncRNAs act as competing endogenous RNA (ceRNA) by sponging miRNA, inhibiting the exchange of miRNA binding elements with mRNA. Several studies have shown that abnormal expression of lncRNAs may be associated with the development of HALI (*Chen et al., 2021a*; *Wu et al., 2022*). For example, *Gadd7* sponges *miR-125a*, thus preventing this miRNA from binding to *Mfn1* (*Liu et al., 2019*), and CASC2 ameliorates HALI by sponging miR-144-3p, preventing this miRNA from binding to AQP1 (*Li et al., 2018*). Those studies highlight the importance of lncRNA in HALI and the potential as biomarkers and therapeutic targets. Our analysis identified 200 differentially expressed lncRNAs involved in HALI development. Combined with the MultiMir database, 148 lncRNA-miRNA-mRNA interactions were found. Among these, we found that lncRNA *H19* interacts with *Sox9*, which may influence AEC-II proliferation and differentiation through *Wnt3a* in the Wnt/$\beta$-catenin pathway under the hyperoxia condition in a lung mouse model (*Wu et al., 2024*; *Xu et al., 2015*). Additionally, the *Sox9* gene has been shown in other lung injury models, such as radiation-induced acute lung injury, to contribute to regeneration *via* the PI3K/AKT pathway, with lung epithelial cells exhibiting increased proliferation potential (*Chen et al., 2021b*). Although not directly verified in HALI mouse models, there is strong experimental evidence supporting our conclusions. Furthermore, *H19* was found to be up-regulated in fibroblasts in idiopathic pulmonary fibrosis, and the H19-Sox9 axis was found to contribute to hepatocyte death and liver fibrosis, both of which were observed in HALI (*Chen et al., 2022*). Therefore, the H19-Sox9 axis may be a potential mechanism through which *Sox9* exerts its reparative effects. However, it is important to note that while correlations between gene expression changes and HALI development have been identified, further research is necessary to establish causative relationships. In addition, there are inherent differences between mice and humans with complex clinical parameters, especially when using animal models to study the pathogenesis of human diseases. Future investigations should confirm the complex mechanisms of post-transcriptional regulation in HALI development and focus on experimental validation of these findings to develop effective interventions for HALI, thereby advancing therapeutic strategies for this condition.

## CONCLUSIONS

This study has provided a comprehensive analysis of the transcriptional and post-transcriptional regulation mechanisms involved in the development of hyperoxia-induced acute lung injury (HALI) using a mouse model. Through transcriptome sequencing and

bioinformatics analysis, 727 differentially expressed genes (DEGs) were identified, with significant enrichment in immune response-related functions, suggesting a correlation between inflammation and the pathogenesis of HALI. Additionally, the study highlights the role of alternative splicing (AS), with 422 genes exhibiting significant AS events, indicating a complex regulatory response to hyperoxic conditions. The construction of a lncRNA-miRNA-mRNA competing endogenous RNA (ceRNA) network identified potential regulatory interactions and biological roles of specific lncRNAs and genes, such as *H19* and *Sox9*, which are associated with the progression of HALI. These findings enhance our understanding of the molecular mechanisms underlying HALI and suggest novel avenues for exploring potential therapeutic targets.

### Funding

This work was supported by the National Natural Science Foundation of China (No. 82303948, No. 81960362), Guizhou Provincial Department of Science and Technology (No: ZK- 2022- 660), and Science and Technology Foundation of Guizhou Health Commission (No: gzwjkj2019-1-068). The Kweichow Moutai Hospital, Shanghai Medical School of Fudan University, and Shanghai Jiao Tong University School of Medicine provided technical, logistical, and financial support. The funders had no role in study design, data collection and analysis, decision to publish, or preparation of the manuscript.

### Grant Disclosures

The following grant information was disclosed by the authors:
National Natural Science Foundation of China: 82303948, 81960362.
Guizhou Provincial Department of Science and Technology: ZK- 2022- 660.
Science and Technology Foundation of Guizhou Health Commission: gzwjkj2019-1-068.
The Kweichow Moutai Hospital, Shanghai Medical School of Fudan University.
Shanghai Jiao Tong University School of Medicine.

### Competing Interests

The authors declare there are no competing interests.

### Author Contributions

- Yundi Chen conceived and designed the experiments, analyzed the data, prepared figures and/or tables, authored or reviewed drafts of the article, and approved the final draft.
- Jinwen Liu analyzed the data, authored or reviewed drafts of the article, and approved the final draft.
- Han Qin performed the experiments, authored or reviewed drafts of the article, and approved the final draft.
- Song Qin conceived and designed the experiments, performed the experiments, prepared figures and/or tables, and approved the final draft.
- Xinyang Huang analyzed the data, prepared figures and/or tables, and approved the final draft.

- Chunyan Wei conceived and designed the experiments, authored or reviewed drafts of the article, and approved the final draft.
- Xiaolin Hu conceived and designed the experiments, analyzed the data, prepared figures and/or tables, authored or reviewed drafts of the article, and approved the final draft.

## Animal Ethics

The following information was supplied relating to ethical approvals (i.e., approving body and any reference numbers):

The care and use of experimental animals were carried out in accordance with the Directory Proposals and approved by committee of the Ethics Committee of Affiliated Hospital of Zunyi Medical College (NO. zyfy-an-2023-0047)

## Data Availability

The datasets generated and analyzed during the current study are available in the Gene Expression Omnibus (GEO) database: https://www.ncbi.nlm.nih.gov/geo/query/acc.cgi?acc=GSE237260.

## Supplemental Information

Supplemental information for this article can be found online at http://dx.doi.org/10.7717/peerj.18069#supplemental-information.

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
