# Peer review of "Deciphering regulatory patterns in a mouse model of hyperoxia-induced acute lung injury"

_PeerJ, doi:10.7717/peerj.18069_

## Round 0.1 · original submission · Major Revisions

To improve the manuscript, please address all the issues raised by the reviewers.

Reviewer 1 ·

Basic reporting

1. Sample Size and Statistical Power (Lines 150):
"In total, 12 samples were included and are evenly separated into 2 groups, with 6 biological replicates on each side. The statistical power of this experimental design, calculated in RNASeqPower is 1.0." While a statistical power of 1.0 seems ideal, it is often unachievable in biomedical research. I suggest providing detailed parameters and the process of how this statistical power was calculated, to justify the adequacy and appropriateness of the sample size chosen.


2 WGCNA Soft Threshold Selection (Lines 261-270):
"Weighted gene co-expression network analysis (WGCNA) is a hierarchical clustering approach employed to categorize genes into discrete expression patterns." Given the sensitivity of WGCNA results to threshold selection, please describe how the soft threshold was determined, whether sensitivity testing for multiple thresholds was conducted, and how these thresholds affect the final network construction.


3 Appropriateness of DESeq2 Parameters (Lines 154-159):
"Significant differentially expressed genes were identified based on the following criteria: log2 fold change > 1.5 or < -1.5 and P-values < 0.05." Please explain why these specific thresholds were chosen, whether they were adjusted based on the distribution of data and outliers, and the potential impact of these parameter settings on the results.

Experimental design

4. Accuracy of ceRNA Network Construction (Lines 195-200):
"The ceRNA network was constructed based on these results and visualized using Cytoscape (v3.10)." When constructing the ceRNA network, please provide detailed information on the miRNA target prediction and validation tools used, the criteria for selecting these tools, and an assessment of the reliability of the results.


5. Quality Control of RNA-seq Data (Lines 133-141):
"Library preparation was performed following the Abclonal mRNA-seq Library Preparation Kit (Abclonal, China) protocol." Given that the quality of RNA-seq data directly affects the reliability of subsequent analyses, detailed descriptions of the quality control steps taken, including how low-quality reads and potential contamination were handled, are recommended.


6. Use of rMATS for Detecting Alternative Splicing (Lines 178-184):
"We employed rMATS (v3.3.0) to quantify the splicing events for each sample and identify differential alternative splicing events." Please specify how significant alternative splicing events were validated and screened, whether consistency analyses across biological replicates were conducted, and the biological interpretation of these splicing events.

Validity of the findings

7. Correction in GO and KEGG Enrichment Analyses (Lines 164-170):
"Enrichment analysis was performed and visualized using the ClusterProfiler package (version 4.2.2)." In performing enrichment analyses, were multiple testing corrections, such as FDR correction, appropriately applied to avoid false positives?


8. Detailed Description of the Experimental Model (Lines 390):
"Mice exposed to >95% oxygen can develop HALI." In describing the experimental model, you mention using a hyperoxic environment to induce HALI, but there is a lack of detailed information about the experimental conditions, such as the specific duration of oxygen exposure and environmental control parameters (like temperature, humidity). Detailed experimental conditions are crucial for the reproducibility of the experiment. Could you provide a more comprehensive description of the experimental setup, including how these conditions comply with relevant animal experiment ethical standards?

Annotated reviews are not available for download in order to protect the identity of reviewers who chose to remain anonymous.

Reviewer 2 ·

Basic reporting

The flow of the manuscript works and the language is professional and clear English. The manuscript is explores the genes that differentially expressed between the two groups HALI vs control, in mouse model. The flow of the article is systematic and nicely explained with some great examples in each results section and the possible consequences of up-regulation and down-regulations.

Experimental design

I commend authors for their extensive data set and analysis. The only comment is have it to add a few more lines on the approach of statistical analysis. Please add a few lines on why was this analysis chosen and add a few lines to describe the same.

Validity of the findings

Conclusions are well stated and supports the aim of the study.

Additional comments

Suggested edit:
Line 426 - please add figure number being discussed here.
Figure 1 - please label the type of cells/lung area being shown in the figure.
Figure 3B - the said colors and labels doesn't match up. Example yellow looks like blue or turquoise and the other way around.

---

## Round 0.2 · Minor Revisions

Please address further comments/suggestions from Reviewer 1.

Reviewer 1 ·

Basic reporting

Clear and Unambiguous, Professional English Used Throughout
The article is written in clear and professional English, but there are a few areas where technical language could be made clearer to enhance readability. Specifically:

In the "Introduction" section, some sentences are complex and could benefit from simplification for better comprehension.
The "Results" section contains several technical terms that could be better explained or referenced to ensure understanding by a broader audience.
Suggested Improvements:

Simplify complex sentences in the "Introduction" to improve readability.
Provide brief explanations or references for technical terms used in the "Results" section.
Literature References, Sufficient Field Background/Context Provided
The article includes an adequate introduction and background, providing sufficient context for the study. However, a few recent studies relevant to the topic are missing and should be included to better position the research within the broader field.

Suggested Improvements:

Include more recent references to studies related to HALI and oxygen therapy to strengthen the background context.
Ensure all referenced literature is appropriately cited in the text.
Professional Article Structure, Figures, Tables. Raw Data Shared
The structure of the article generally conforms to standard scientific reporting formats. The figures and tables are relevant and appropriately labeled, but a few figures lack sufficient resolution, which may hinder detailed examination.

Suggested Improvements:

Ensure all figures are of sufficient resolution for detailed examination.
Consider reformatting some sections to align more closely with standard scientific reporting conventions if they improve clarity.
Self-Contained with Relevant Results to Hypotheses
The article is self-contained and presents results that are relevant to the stated hypotheses. All appropriate raw data have been made available in accordance with the Data Sharing policy.

Suggested Improvements:

No significant improvements needed in this area as the article already meets the standards well.
General Comments:
Overall, the article is well-written and meets the majority of the journal's basic reporting standards. Addressing the above points will further improve clarity, comprehensiveness, and alignment with professional standards.

Experimental design

Original Primary Research within Aims and Scope of the Journal
The research is well within the aims and scope of the journal, focusing on deciphering regulatory patterns in a mouse model of hyperoxia-induced acute lung injury (HALI).

Research Question Well Defined, Relevant, & Meaningful
The research question is clearly defined, relevant, and meaningful, addressing the mechanisms underlying HALI. The study identifies a significant knowledge gap in understanding the transcriptional regulation in HALI and aims to fill this gap through comprehensive analysis.

Suggested Improvements:

Although the research question is clear, a more detailed explanation of how this study builds on or differs from previous work could strengthen the introduction.
Rigorous Investigation Performed to a High Technical & Ethical Standard
The investigation is conducted rigorously, adhering to high technical and ethical standards. The study follows established protocols for animal research and ensures compliance with ethical guidelines.

Suggested Improvements:

Consider providing more detailed information on the ethical approval process and any specific measures taken to ensure animal welfare during the study.
Methods Described with Sufficient Detail & Information to Replicate
The methods section is detailed and provides sufficient information for replication. The descriptions of the experimental procedures, RNA sequencing, and data analysis are thorough and clear.

Suggested Improvements:

Ensure that all reagents, software, and equipment used are fully detailed, including model numbers and sources, to facilitate exact replication by other researchers.
Consider adding a flowchart or diagram summarizing the experimental design to provide a quick visual reference for the methodology.
General Comments:
Overall, the experimental design is well-executed and clearly presented, meeting the journal's standards for originality, relevance, and methodological rigor. Addressing the suggested improvements will enhance the clarity and reproducibility of the study.

Validity of the findings

Validity of the Findings Review Comments
Impact and Novelty Not Assessed, Meaningful Replication Encouraged
The study presents original research within the scope of the journal, addressing a significant issue in hyperoxia-induced lung injury. The research contributes to existing knowledge by detailing specific gene regulatory mechanisms and interactions, which have not been extensively covered in previous studies. However, the article could better emphasize how the findings extend current knowledge or introduce novel methodologies.

Suggested Improvements:

Clearly state the novel aspects of the study and how these findings expand upon or differ from previous research.
Discuss the potential implications of the research in broader terms and how it could influence future studies or clinical practices.
All Underlying Data Provided; Robust, Statistically Sound, & Controlled
The study provides all relevant data, which appears to be robust and statistically sound, adhering to standard practices for data sharing and transparency. However, details on the control measures used during experiments and the statistical methods could be more thoroughly explained to ensure the robustness of the findings.

Suggested Improvements:

Enhance the description of statistical methodologies and control measures to bolster confidence in the robustness of the data.
Ensure that all data, including raw and processed data, are available in a recognized discipline-specific repository with clear access instructions.
Conclusions Well Stated, Linked to Original Research Question & Limited to Supporting Results
The conclusions are well articulated and directly related to the research question. They are supported by the results presented. However, some conclusions may benefit from a more cautious presentation to avoid overstating the causative relationships suggested by the findings.

Suggested Improvements:

Refine the language used in the conclusions to clearly differentiate between correlation and causation, especially in the context of gene expression and lung injury responses.
Provide a more detailed discussion on the limitations of the study and how these might affect the interpretation of the results.
General Comments:
The validity of the findings is generally well-supported by the data and analysis presented. Enhancing the clarity of the novel contributions, ensuring complete transparency of data, and carefully stating the conclusions will improve the credibility and impact of the study.

Additional comments

This manuscript presents a comprehensive study on hyperoxia-induced acute lung injury (HALI) in a mouse model, exploring complex gene regulatory networks and potential therapeutic targets. The research is thorough, with rigorous experimental design and a robust analysis. Here are some additional considerations and suggestions that could further enhance the manuscript:

Interdisciplinary Relevance: The study delves deep into the genetic mechanisms but could also discuss the broader implications for clinical practice and potential therapeutic interventions. Linking the findings more explicitly to possible clinical outcomes could greatly increase the relevance and impact of the work.

Comparative Analysis: While the manuscript effectively identifies and analyzes differentially expressed genes and pathways, a comparison with other models or conditions of lung injury could contextualize the findings within the larger field of pulmonary research. This comparative analysis could highlight the unique or particularly significant aspects of HALI.

Long-term Effects and Reversibility: The study primarily focuses on the immediate genetic responses to hyperoxia. Including or discussing potential long-term effects and the reversibility of the observed changes upon removal of the hyperoxia stimulus could add valuable insights into the progression and potential recovery from HALI.

Limitations: While some limitations are briefly mentioned, a more detailed discussion would strengthen the manuscript. Specifically, addressing the translatability of mouse model findings to human conditions and any methodological constraints would provide a more balanced view of the study’s implications.

Future Research Directions: Suggestions for future studies based on the findings could guide subsequent research in this area. This might include investigating specific gene targets for therapy or exploring preventative strategies against hyperoxia-induced damage.

Graphics and Visual Enhancements: Some of the figures, while informative, could be enhanced with additional graphical elements or interactive features (if the journal format allows), which could make the data more accessible and engaging to readers.

Overall, the manuscript is a valuable contribution to the field of respiratory research and offers significant insights into the genetic underpinnings of HALI. Addressing these additional comments could make the paper even more comprehensive and impactful.

---

## Round 0.3 · Minor Revisions

Please address the final suggestions of reviewer 1

Reviewer 1 ·

Basic reporting

no comment

Experimental design

no comment

Validity of the findings

no comment

Additional comments

Dear Authors,

Thank you for submitting your manuscript titled "Deciphering Regulatory Patterns in a Mouse Model of Hyperoxia-Induced Acute Lung Injury" and for addressing the initial review comments. Your revisions have significantly enhanced the quality of the manuscript. As you prepare the final submission, please consider the following suggestions to further refine the completeness and depth of your work:

Language and Expression: The overall clarity of the manuscript is commendable, however, there are instances of inconsistent terminology, particularly in Section 3, where terms like "upregulation" and "activation" are used interchangeably. These terms do not always carry the same biological implications.

Suggested Improvement: I recommend unifying the use of technical terms and clearly defining them to ensure accuracy and professionalism in scientific expression. This will aid comprehension not only for experts but also for readers less familiar with the field.
Recent Literature Citations: While the introduction covers a broad range of background knowledge on gene regulation under hyperoxia, it lacks citations from key studies published after 2022.

Suggested Improvement: Please incorporate references from key research published in the last year or two, especially those revealing new findings about the role of long non-coding RNAs in lung injury. This will demonstrate the timeliness and depth of your research.
Optimization of Figures: Figure 5, depicting the ceRNA network, although detailed, has labels and connecting lines that are overly congested, which compromises the clarity of information conveyed.

Suggested Improvement: Consider simplifying the figure by using different colors or symbols to differentiate between types of RNA and proteins, and enhance the legend for clearer understanding. This will ensure the information is immediately graspable.
Details on Data Sharing: You have done well to share your experimental data in the GEO database. However, the manuscript lacks detailed descriptions of how these data can be accessed and utilized by others.

Suggested Improvement: Describe in detail in the methods section how the data can be accessed and used, providing specific database links and dataset identifiers to ensure that other researchers can easily replicate and validate your experimental results.
By refining these details, your research will offer richer and more precise scientific insights to peer reviewers. I look forward to your revised submission.

Best regards,

---

## Round 0.4 · accepted · Accept

The authors have satisfactorily addressed the last issues raised by reviewer 1.
Before publication, please recheck the text (line 43: "from" and not "form"; line 277: "that" should be omitted).